# Techno-Stress Creators, Burnout and Psychological Health among Remote Workers during the Pandemic: The Moderating Role of E-Work Self-Efficacy

**DOI:** 10.3390/ijerph20227051

**Published:** 2023-11-10

**Authors:** Chiara Consiglio, Nicoletta Massa, Valentina Sommovigo, Luigi Fusco

**Affiliations:** Department of Psychology, Faculty of Medicine and Psychology, Sapienza University of Rome, Via dei Marsi 78, 00185 Rome, Italy; nicoletta.massa@uniroma1.it (N.M.); valentina.sommovigo@uniroma1.it (V.S.); luigi.fusco@uniroma1.it (L.F.)

**Keywords:** remote working, technostress, e-work self-efficacy, COVID-19 pandemic, burnout assessment tool, psychological health

## Abstract

During the COVID-19 pandemic, remote working was pervasively implemented, causing an increase in technology-related job demands. Concurrently, there was an increase in psychological problems in the occupational population. This study on remote workers tested a moderated mediation model positing burnout, conceptualized according to the Burnout Assessment Tool, as the mediator between techno-stressors and psychological health outcomes and e-work self-efficacy as a protective personal resource. A sample of 225 remote workers filled out anonymous questionnaires measuring techno-stressors, e-work self-efficacy, burnout, and psychological health symptoms (i.e., depressive mood and anxiety symptoms). The data were analyzed using structural equation mediation and moderated mediation models, adopting a parceling technique. The results showed that burnout totally mediated the relationship between techno-stressors and depressive mood, while partially mediating the association between techno-stressors and anxiety symptoms. Moreover, e-work self-efficacy buffered the positive effects of techno-stressors on depressive mood and anxiety symptoms through burnout. The present research attested to the relevance of techno-stressors for the psychological health of remote workers and supported burnout as a mediator of this process, although anxiety symptoms were also directly related to techno-stressors. Moreover, the protective role of domain-specific self-efficacy was confirmed in the realm of remote working. Limitations and practical implications are discussed.

## 1. Introduction

Over the last decade, the diffusion of Information and Communication Technology (ICT) within organizational contexts has exponentially increased, especially following the COVID-19 outburst [1]. Indeed, during the pandemic, governments took restrictive measures in the labor sector, which forced organizations to close their offices and adopt new work arrangements that replaced face-to-face activities with remote working solutions. Such a change pushed millions of employees worldwide to face the radical reshaping of the work setting [2,3], which became fully based on new technologies [4].

Although working remotely is often associated with a greater sense of autonomy, flexibility, and job satisfaction [5,6,7], it may also have detrimental effects on employees’ health, job performance, and work–life balance [8,9,10,11,12]. Specifically, employees who worked remotely during the pandemic often experienced the sudden displacement of their activities from office to home, and from a real to a virtual work setting, without the possibility of flexibly organizing time, spaces, and work mode that usually come along with remote working [13]. Additionally, most people were not adequately trained to move to this new way of working, making them more exposed to additional job demands related to the use of technologies and to the management of their online work role [14]. Accordingly, remote workers were exposed to increasing techno-stressors (or technostress creators), such as the difficulty of learning how ICT systems work (i.e., techno-complexity), the intensification of overload and work pace (i.e., techno-overload) and the invasiveness of work within one’s private life due to technology (i.e., techno-invasion). Moreover, techno-stressors may have harmful effects in terms of performance loss and psycho-physical distress [15,16,17,18].

At the same time, the pandemic context was also characterized by an increase in several psychological health symptoms related to anxiety and depression [19,20,21]. Nevertheless, only a few studies have focused on how technological stressors may relate to these mental health outcomes [22,23], and the process through which this may occur is still poorly understood. Given the spread of the phenomenon and the intention of many organizations to keep remote or hybrid work as a stable way of working, more research is needed to reach a deeper understanding of how remote working techno-stressors relate to psychological health outcomes, and which factors can protect against such detrimental effects. Indeed, remote working requires people to develop both digital (e.g., learning how to properly use ICT systems) and soft skills (e.g., emotional and social regulation) [24]. For this reason, perceiving oneself as being able to handle the challenges associated with working remotely (e.g., managing work and private boundaries appropriately and maintaining a positive working relationship with colleagues and the manager while working remotely) represents a valuable resource in this new working scenario. Indeed, self-efficacy in the context of remote work (i.e., e-work self-efficacy [25]) enables workers to better cope with techno-stressors and then protect them from developing job stress-related detrimental outcomes [16,26,27,28]. Although the protective role of self-efficacy against burnout and mental health problems is well-known [29], this is the first study to investigate e-work self-efficacy as a personal boundary condition explaining when the effects of techno-stressors can translate into remote workers’ burnout and mental health problems. Investigating the protective role of e-work self-efficacy is important because it provides new practical insights into how practitioners can support remote workers’ mental health by providing tailored training programs for the development of their digital resilience.

In the present study, we refer to the Conservation of Resources Theory (COR-Theory) [30] as our main theoretical framework. Indeed, the COR theory assumes that people “strive to retain, protect, and build resources and that what is threatening to them is the potential or actual loss of these valued resources” [30] (p. 516). Nevertheless, when people perceive that they do not have adequate resources to face high job demands (i.e., techno-stressors), they enter a defensive mode, becoming more vulnerable to further resource losses and impaired psychological health [31]. Indeed, to preserve their well-being in challenging circumstances, people must invest resources to gain well-being. Therefore, employees who can rely on more personal resources (i.e., e-work self-efficacy) will be more likely to balance work demands while preserving their health and gaining additional protective resources. Accordingly, the aims of the present research are threefold:(1)To explore whether techno-stressors are related to burnout, measured using the Burnout Assessment Tool [32];(2)To examine whether burnout mediates the relationship between techno-stressors and psychological health outcomes (i.e., anxiety symptoms and depressive moods);(3)To investigate whether the detrimental effects of techno-stressors on anxiety symptoms and depressive mood through burnout are buffered by e-work self-efficacy levels.

Figure 1 represents the conceptual model hypothesized in the present study. In the following paragraphs, the specific literature and hypotheses are presented.

### 1.1. From Remote Working Techno-Stressors to Burnout

The organizational advantages of new working arrangements (i.e., remote working) are well known nowadays [33,34,35]. Nevertheless, working remotely exposes employees to some ICT-related job features (i.e., techno-stressors) that may represent additional job demands potentially affecting their well-being [28,36]. In this regard, Tarafdar and colleagues [9] identified several techno-stressors that represent inherent aspects of ICT systems, potentially associated with strain outcomes. Among them, techno-complexity refers to those ICT characteristics that make users feel inadequate regarding their digital skills. Indeed, difficulty in effectively using new technologies may be associated with employees’ higher levels of frustration and stress [37]. Techno-invasion refers to the invasiveness that technology brings outside of work, leading a person to always stay connected to job-related tasks. Consequently, work and private dimensions become increasingly intertwined, and employees may find it more difficult to detach from work and adequately manage different role demands [38], thus negatively affecting their well-being [39]. Techno-overload relates to the ICT potential to compel people to work faster and longer or to change work habits due to overexposure to information channeled by technologies (e.g., emails, and notifications) [40,41]. Indeed, the struggle to cope with different streams of information has been associated with an individual’s increased vulnerability to burnout and exhaustion [42]. All in all, the technostress literature suggests that the prolonged and mandatory use of ICT tools may deplete personal resources and have a detrimental effect on an individual’s psycho-physical health (e.g., [43]). For instance, a previous study on this topic [44] showed that interacting with technology may lead people to feel exhausted and anxious. Moreover, both cross-sectional (e.g., [45]) and diary studies (e.g., [46]) have highlighted how the constant need to update one’s digital skills, as well as to manage the work–family boundaries that technology blurs, makes people more exposed to harmful psychological and health outcomes. In this regard, the negative effects of techno-stressors on remote workers’ psychological health were especially evident in the pandemic context, where the job resources required to optimize remote working (e.g., adequate technical support, specific training for employees and managers) were often insufficient to face such a sudden change [2,6]. Indeed, Molino and colleagues [47] found a positive relationship between techno-stress creators and work-related stress during the first Italian pandemic lockdown. Furthermore, a recent narrative review [48] showed that employees who worked remotely during the pandemic experienced higher workload levels due to technologies, greater difficulty in effectively demarcating between work and home settings, and poor mental health.

Burnout, which is the most studied occupational stress-related syndrome, refers to individuals’ negative responses to chronic work-related stress [49]. This phenomenon may occur in a variety of work settings [50] and may be driven by a wide range of job demands, including techno-stressors [16,17,45,51,52,53]. A large part of the literature has operationalized burnout as a syndrome comprising three specific dimensions, namely emotional exhaustion (i.e., the feeling of having drained all psychological energies), cynicism (i.e., the mental detachment toward work), and reduced professional efficacy (i.e., the perceived lack of professional accomplishment) [54]. This conceptualization of burnout is detectable through the Maslach Burnout Inventory [55]. Even if, to date, the MBI is the most widely used tool to assess burnout, this instrument has some methodological and practical limitations [56]. Specifically, these limitations are attributable to issues related to the factorial validity [57,58,59] and factorial invariance of this scale [60], as well as the impossibility of combining its three dimension scores (i.e., emotional exhaustion, cynicism, and reduced accomplishment) in a single burnout total score [61]. As a result, most studies using the MBI to measure burnout consider it the product of three dimensions that cannot be combined. Consequently, often the role of the key dimension of burnout, namely exhaustion, has been overemphasized, e.g., [62], underestimating the other burnout dimensions, e.g., [62,63]. An alternative proposal recently came with the development of the Burnout Assessment Tool (BAT) [32]. Specifically, the authors developed a new conceptualization of burnout that comprises four interrelated core dimensions. Among these, two core burnout aspects were already present in the MBI, namely the energetic (exhaustion) and motivational (cynicism) components [51]. According to this view, exhaustion refers to physical and psychological resource depletion, while mental distance describes disengagement toward one’s work and disenchantment with its meaning. Moreover, through interviews of healthcare professionals working with burned-out employees, two additional recurrent symptoms were identified and added to the original core dimensions [32], namely emotional impairment (i.e., the difficulty in regulating one’s emotional processes) and cognitive impairment (i.e., the loss of self-regulation of one’s cognitive processes). An important novelty of this approach to burnout compared to the MBI is the possibility of combining the four-dimension scores in a burnout total score.

In line with the Conservation of Resources Theory [30], working with technology requires employees to invest their resources to manage increasing ICT-related demands. This effort may activate a spiral of resource losses [31,64], eventually resulting in burnout symptoms. Indeed, given that remote workers are continuously exposed to technology-related demands, such as techno-complexity, techno-overload, and techno-invasion, they may experience the feeling of having worn out their own energetic and psychological resources (i.e., exhaustion). The resulting resource depletion may elicit negative emotions and affect their capability to invest additional resources to control cognitive and emotional processes (i.e., cognitive, and emotional impairment). Moreover, based on the desperation principle of COR theory [31], resource-depleted employees may also adopt a defensive attitude while trying to keep work at a distance to protect their residual resources (i.e., mental distance). Hence, we hypothesize:

**Hypothesis** **1:**
*Techno-stress creators will be positively related to burnout.*


### 1.2. Techno-Stressors, Burnout, and Psychological Health Outcomes

Based on the Burnout Assessment Tool [32], burnout represents a multifaceted experience, where exhaustion and disengagement from one’s work are compounded by the weakening of control over cognitive and emotional processes (e.g., having trouble focusing or controlling one’s negative emotional states at work), thus making the person more susceptible to developing physical and psychological problems [32]. A previous study [65] found that burned-out individuals were more likely to experience anxiety symptoms. Similarly, another study [21] reported that the more people perceived themselves as burned-out, the more they felt anxious. Additionally, although there is still a debate about the distinctiveness or possible overlapping between burnout and depression [66,67], many studies have investigated the relationship between the two concepts, as distinct phenomena [68,69,70]. For instance, a three-wave prospective study [71] found that burnout predicts depressive symptoms. Furthermore, there is longitudinal evidence that an increase in burnout over time is predictive of an increase in depressive symptomatology over time [72,73]. However, most studies on the relationships of burnout with depressive mood and anxiety symptoms analysed burnout as measured by the MBI, mainly focusing on the emotional exhaustion dimension, and underestimating the other burnout components. To date, only a few studies have shown positive correlations of burnout as measured by the BAT, with depressive mood [74] and anxiety symptoms [75], with no previous research on remote workers during the pandemic.

Consistent with this line of research and according to COR theory principles [31], when people’s resources are exhausted, they are more likely to end up in a resource loss cycle, experiencing a gradual and consistent depletion of energetic and psychological resources. Specifically, the dysregulation of cognitive and emotional processes that characterize burnout syndrome, as it is conceptualized according to the Burnout Assessment Tool [32], may imply an increase in negative thoughts, worries, and rumination, resulting in a worsening of mood and an increase in anxiety symptoms [76,77]. Thus, we hypothesized:

**Hypothesis** **2a:**
*Burnout will be positively related to depressive mood.*


**Hypothesis** **2b:**
*Burnout will be positively related to anxiety symptoms.*


Previous research has empirically demonstrated that techno-stressors are negatively related to mental health [78]. Thus, when confronted with techno-related demands, employees may experience lower well-being levels, which may activate a vicious cycle of resource depletion [79]. Indeed, failing to handle the complexity of technology, combined with struggling to set work priorities and boundaries between work and private life, makes the person less able to cope with burnout symptoms, e.g., [18]. This, in turn, may make employees even more vulnerable to the development of a depressive mood and anxiety symptoms [80,81]. In other words, constantly dealing with a technologically demanding environment can gradually weaken individuals’ energetic, motivational, emotional, and cognitive resources. Thus, techno-stressed workers are more likely to develop burnout, which may, in turn, exacerbate anxiety and depressive mood symptoms. Although there is some preliminary evidence of correlations between BAT and mental health symptoms (including depression and anxiety symptoms [74,75]), this is the first study to empirically investigate whether the total burnout score, as measured by the BAT, can be a mechanism for translating the effects of techno-stressors in burnout and depressive symptoms. As such, and given the mediational model of burnout [82], we propose:

**Hypothesis** **3a:**
*Burnout will mediate the relationship between techno-stress creators and depressive mood.*


**Hypothesis** **3b:**
*Burnout will mediate the relationship between techno-stress creators and anxiety symptoms.*


### 1.3. E-Work Self-Efficacy as a Protective Personal Resource

Drawing on the COR theory [30], people are motivated to maintain, promote, and protect resources for future needs and mitigate stressful experiences. Specifically, individuals with greater resources will be more likely to perceive the demands of the environment to a lesser amount and, at the same time, to take advantage of their personal and environmental resources to overcome job demands [31]. Among personal resources, self-efficacy is the most powerful in influencing people’s behavior and well-being [29,83,84,85], shaping personality functioning through its influence on emotions, thoughts, and motivation [86]. Self-efficacy refers to individuals’ beliefs in their ability to successfully accomplish a particular task, effectively deal with specific situations, or achieve specific goals [87]. Therefore, it underlines the importance of human agency in empowering one’s competencies, overcoming self-doubts, and managing change, uncertainty, and challenging situations [88,89,90]. A further distinctive feature of self-efficacy is its domain-specific nature [91]. This makes the construct particularly suitable for investigating specific realms of experience, capturing the relevance of personal beliefs in different domains or contexts. In the case of remote working, the employee is challenged by a varied range of technology-related job demands (i.e., techno-stressors), which involve not only the effective use of technological tools but also the deployment of relational and role skills. Indeed, although previous studies in the IT work domain mostly related self-efficacy beliefs with the perceived competence to successfully perform computer-related tasks, as in the case of computer self-efficacy [92,93], there is now a growing awareness of the importance of a more comprehensive view on the different capabilities needed to manage and optimize the remote working experience [25,27]. Within this new work arrangement, employees need to plan and organize their work and priorities, set work–life boundaries, define their job responsibilities [94], reshape work collaboration in a virtual setting [95], and increase their ICT-related knowledge [8]. This requires a certain degree of proactivity, self-organization, and awareness of one’s own resources in dealing with technology-related demands and with the job autonomy that comes with working remotely. Consequently, self-efficacious people are more likely to invest personal resources, acting in the environment even in the presence of potential obstacles [96,97]. Overall, this may represent a protective factor against psychological health impairment [98]. This may also reduce the risk that continuous ICT usage may lead to burnout [99]. In fact, according to previous research and in line with the COR theory principles, we argue that people who are high in e-work self-efficacy will be more capable of and effective in dealing with techno-stressors, being less vulnerable to resource losses, and less likely to experience the detrimental effects of techno-stressors on their psychological well-being. Thus, we hypothesized:

**Hypothesis** **4:**
*E-work self-efficacy will moderate the relationship between technostress creators and burnout so that when facing technostress creators, employees with higher e-work self-efficacy will be less likely to experience burnout symptoms and consequently depressive mood and anxiety symptoms than those with lower e-work self-efficacy.*


Moreover, to ascertain the distinctive contribution of e-work self-efficacy as a protective individual resource for remote workers, we are also interested in comparing its moderating effect with that played by another personal resource, namely resilience. Resilience represents the capacity to bounce back in the face of adversities, failures, and crises to regain personal balance [100]. Resilience has been identified by the COR theory as an important protective individual factor [101]. Therefore, as an additional exploratory research question of the present study, we will also examine the moderating role of resilience in the relationship between techno-stressors and burnout.

## 2. Materials and Methods

### 2.1. Participants and Procedure

This study was conducted in Italy between June and October 2020 during the COVID-19 pandemic. The survey was carried out through the compilation of an anonymous self-report questionnaire using the Qualtrics platform. The link to invite participants to complete the survey was distributed through social networks (i.e., LinkedIn) and personal contacts. All procedures performed in this study were in accordance with the ethical standards of the 1964 Helsinki Declaration and its later amendments or comparable ethical standards. Data storage met current Data Protection regulations. The questionnaire’s cover sheet informed the participants about the study’s goal and ensured both the voluntariness of their participation and the confidentiality of the responses. The inclusion criteria were as follows: to be at least 18 years of age, to be working remotely, and to provide an explanation of the research goals and an informed consent form. Participation in the study was voluntary, and the team guaranteed complete anonymity to all respondents. No personal data were collected for the research. The resulting sample was composed of 349 remote workers. We excluded 124 subjects because of incomplete responses (i.e., less than 50% of the questionnaire). Thus, the final sample consisted of 225 remote workers from several organizations and sectors. Most of them were women (60%), aged between 26 and 35 years (35.6%), followed by workers 46–55 years old (20%), 18–25 years old (16.9%), 36–45 years old (16.9%), and over 55 years old (10.7). Regarding the job, 32.9% of the respondents were office workers, followed by highly specialized professionals (28.4%). More than half of them had a permanent contract (56.4%). Most of the respondents had a university degree (48.9%), followed by a post-graduate degree (23.1%) and a high school diploma (27.6%). Most of the participants did not have any children (61.3%), followed by those who had two children (19.1%), one child (13.3%), three children (5.3%), and more than three children (0.9%).

### 2.2. Measures

#### 2.2.1. Techno-Stressors

Techno-stressors were adapted by the Technostress Creators scale [40], which measures three dimensions: techno-overload (5 items, e.g., “I am forced by technology to work much faster”; α = 0.93), techno-invasion (4 items, e.g., “I have to be in touch with my work even during my vacation due to this technology”; α = 0.86), and techno-complexity (5 items, e.g., “I need a long time to understand and use new technologies”; α = 0.91). Respondents reported how frequently they experienced technostress situations on a five-point Likert scale (1 = “strongly disagree”, 5 = “strongly agree”). The total techno-stressors scale reliability was α = 0.94.

#### 2.2.2. Burnout

Burnout was assessed using the Italian version of the short version of the Burnout Assessment Tool [102,103,104]. This scale includes the following four dimensions: exhaustion (3 items, e.g., “At work, I feel mentally exhausted”), mental distance (3 items, e.g., “I struggle to find any enthusiasm for my work”), reduced emotion control (3 items, e.g., “At work, I feel unable to control my emotions”), and reduced cognitive control (3 items, e.g., “At work, I have problems staying focused”). Respondents reported how frequently they experienced each of the listed symptoms on a 5-point Likert scale (1 = “never”, 5 = “always”; α = 0.92).

#### 2.2.3. Depressive Mood and Anxiety Symptoms

Depressive mood and anxiety symptoms were measured using the short version of the Depression Anxiety Stress Scales (DASS-21) [105]. Participants indicated how frequently during the COVID-19 pandemic they had experienced each of the depressive moods (e.g., “Feeling discouraged and depressed” α = 0.81) and anxiety symptoms (“Feeling close to having a panic attack” α = 0.82) listed on a 4-point Likert-type scale (1 = “it never happens to me”, 4 = “it happens to me most of the time”).

#### 2.2.4. E-Work Self-Efficacy

E-work self-efficacy was measured by adapting five items from an e-work self-efficacy scale [25] for the specific purpose of this study. Respondents were asked to report their beliefs about their own capability to manage the challenges of working remotely, using the following items: “I am able to meet all the deadlines in my job even working remotely”; “Working remotely, I am able to effectively balance family and work commitments”; “I am able to maintain close relationships with colleagues even when working remotely”; “I can be appreciated by my supervisor even when working remotely”; “I am able to actively participate in meetings even when working remotely”) on a 7-point Likert scale (1 = “completely false for me”, 7 = “completely true for me”; α = 0.77).

#### 2.2.5. Resilience

Resilience was assessed by adapting four items taken from the resiliency scale of the Psychological Capital Questionnaire [106], specifically reworded to refer to the pandemic crisis. Respondents were asked to report their perceived capacity of facing, bouncing back from the pandemic crises, and regaining personal balance (e.g., “I think I can easily get through this moment of pandemic crisis”; α = 0.82) on a 5-point Likert scale (1 = “totally disagree”, 5 = “totally agree”).

### 2.3. Statistical Analysis

Using IBM SPSS Statistics 25, we first examined the psychometric properties of the study scales. We considered values of the variance inflation factor and tolerance below the recommended threshold of 10 as indicators of no sign of multicollinearity. The normality of the data was considered acceptable when skewness < |3.0| and kurtosis < |8.0| [107]. To test the construct’s reliability, we examined individual item reliability (i.e., items with factor loadings on their respective constructs higher than the 0.5 thresholds were considered acceptable) [108], Composite Reliability (CR, i.e., CR > 0.70 was considered acceptable) [109], Average Variance Extracted (AVE, i.e., AVE > 0.50 was deemed acceptable) [110], Cronbach’s alpha (α), and McDonald’s omegas (ω, i.e., internal reliability > 0.70 was deemed acceptable) [111]. Then, we explored correlations among the study constructs using (a) Pearson’s correlation coefficients to measure the strength of the linear link between continuous variables; (b) Spearman’s rho correlation coefficients to test the correlation between ordinal variables or between ordinal/continuous and dichotomous nominal variables; and (c) Kendall’s coefficients of rank correlation tau-subb to evaluate the relationship between continuous and ordinal variables. Next, we used structural equation modeling to examine the appropriateness and fit of our expected conceptual models. Given the limited size of our sample (i.e., 225), taking into account the large number of items, we adopted the parceling technique to maintain an optimal indicator-to-sample-size ratio [112]. Thus, model fit indices may become problematic when the subject-to-item ratio is below the recommended 10:1 ratio [107], as in the case of our research. According to what has been proven [111,112], item parcels ameliorate the sample size-to-parameter ratio, reducing the odds that parcels will be affected by the method effects related to single items. Moreover, item parcels are more reliable, as they reflect a broader proportion of true-score variance and increase convergence and stability, being especially suitable for models having an unfavorable indicator-to-sample-size ratio [113]. Following these considerations [114], we then created three parcels for the key measures by combining items with higher factor loadings with those with lower factor loadings.

Then, we performed confirmatory factor analyses (CFAs) with the maximum likelihood (ML) method to compare our expected measurement model with a series of alternative models in Mplus Version 8. In line with the technostress literature, e.g., [9,99], we tested a second-order structure for techno-stress creators. The model goodness-of-fit of the CFAs and subsequent mediation models were estimated based on the following fit indices: the ratio of the χ^2^ statistic to the respective degrees of freedom (χ^2^/df, values ≤ 2 indicates a superior fit between the hypothesized model and the sample data [114], the root mean squared error of approximation (RMSEA, values less than 0.08 and 0.05 suggest an adequate and good model fit) [115]; the standardized root mean square residual (SRMR; values less than 0.08 are taken as a good fit) [115]; the comparative fit index (CFI, values above 0.90 are indicative of a good model fit) [115]; and the Tucker–Lewis index (TLI, values between 0.90 and 0.95 indicate acceptable fit) [116]. After establishing a good fit for the measurement model, we ran Harman’s single-factor test and the unmeasured latent method factor technique [117] to assess the potential impact of common method bias on our study (the single factor should explain less than 50% of the variance, and the method factor should explain equal to or less than 25% of the method variance, which is the average amount of method variance referred to in self-reported research [117].

After establishing a good fit for the measurement model, to test the hypothesized mediating role of burnout in the relationship between techno-stressors and its expected two outcomes (i.e., depressive mood, and anxiety symptoms), we conducted a mediation analysis with the Maximum Likelihood (ML) method in Mplus Version 8, using the bootstrapping test and a bias-corrected 95% confidence interval (CI) with a resampling procedure of 1000 bootstrap samples. This method is especially suitable for estimating indirect effects because it provides non-symmetric confidence intervals [118]. Moreover, bias-corrected bootstrap analysis is recommended as it is less biased and more powerful compared to alternative bootstrap methods [119]. Mediation analyses were conducted while controlling antecedents, mediators, and outcomes for gender, age, educational level, and number of children, in line with previous literature [8,101,120]. We also confirmed our results by testing a mediation model with the exclusion of covariates. Additionally, we compared the fit indices of our hypothesized mediation model with those of the reverse mediation model. Next, to examine whether the strength of the association between techno-stressors and expected outcomes (i.e., depressive mood, and anxiety symptoms) through burnout was conditional on the values of e-work self-efficacy, we conducted a moderation mediation model using Mplus Version 8, while controlling antecedent, mediator, and outcomes for gender, age, educational level, and number of children. We evaluated the goodness of our expected moderated meditation model by comparing it in terms of the Bayesian Information Criterion (BIC), and Akaike Information Criterion (AIC) with an alternative model using resilience as a moderator. Indirect and conditional effects were considered statistically significant when the 95% confidence interval (CI) from the bootstrap analysis did not include zero and the *p*-value was less than or equal to 0.05. Lower AIC and BIC values indicated a better fit, so that the best-fitting model was that with the lowest AIC and BIC values. Finally, to plot the statistically significant interaction effects, we used specific Excel worksheets that allowed us to calculate these effects [121].

## 3. Results

### 3.1. Measurement Reliability of Study Constructs and Descriptive Analyses

There was no sign of multicollinearity given that the variance inflation factor values ranged from 1.28 to 1.48, and the tolerance values ranged from 0.67 to 0.76. The values of skewness (ranging from −0.74 to 1.08) and kurtosis (ranging from −0.54 to 1.48) were appropriate. All factor loadings of items on their corresponding constructs were statistically significant, suggesting at least a medium correlation with their respective constructs (i.e., techno-stressors: 0.57–0.82; burnout: 0.61–0.82; depressive mood: 0.83–0.93; anxiety symptoms: 0.67–0.81; e-work self-efficacy: 0.69–0.79). Moreover, the results showed that the CR coefficients for the study variables ranged from 0.85 to 0.94. All AVE values for the study variables ranged from 0.52 to 0.66. All study scales had satisfactory internal consistencies showing α values ranging from 0.77 to 0.94 and ω values ranging from 0.77 to 0.94. The descriptive statistics, internal reliabilities, and correlations for the study variables are reported in Table 1.

Techno-stressors were positively related to burnout (r = 0.43, *p* < 0.01), depressive mood (r = 0.29, *p* < 0.01) and anxiety symptoms (r = 0.44, *p* < 0.01), while negatively associated with e-work self-efficacy (r = −0.30, *p* < 0.01). Burnout was positively related to depressive mood (r = 0.48, *p* < 0.01) and anxiety symptoms (r = 0.36, *p* < 0.01), while negatively associated with e-work self-efficacy (r = −0.34, *p* < 0.01). All correlations among the study constructs were in the expected direction.

### 3.2. Confirmatory Factor Analysis and Common Method Bias Check

The fit indices of the five-factor model having two second-order factors (i.e., techno-stressors, and burnout) and three parcels for each construct were satisfactory (χ^2^ = 565.03, df = 388, *p* = 0.00, RMSEA = 0.04, RMSEA [90% CI] = [0.04, 0.05], SRMR = 0.05, CFI = 0.96, TLI = 0.95). This model outperformed all alternative models, supporting the distinctiveness of the study variables (see Table 2). The results of Harman’s single-factor test [122] indicated that the first factor explained only 29.91% of the variance. Then, no single factor had particularly significant exploratory power. Additionally, the unmeasured latent method factor explained only 6.94% of the total variance (less than 25% of the average amount of method variance observed in self-report studies) [117], suggesting that common method bias is unlikely to be a major concern.

### 3.3. Hypothesis Testing

In our expected mediation model (χ^2^ (466) = 805.49, χ^2^/df = 1.73, CFI = 0.92; TLI = 0.91, RMSEA = 0.06; 90% CI RMSEA = [0.05, 0.06], SRMR = 0.08; see Table 3), techno-stressors were positively related to burnout (β = 0.47, *p* < 0.001, 95% CI [0.34, 0.61) which, in turn, was positively related to depressive mood (β = 0.44, *p* < 0.001, 95% CI [0.28, 0.62]) and to anxiety symptoms (β = 0.22, *p* < 0.01, 95% CI [0.08, 0.36]), confirming Hypotheses 1, 2a and 2b. Techno-stressors were positively and directly related to anxiety symptoms (β = 0.38, *p* < 0.001, 95% CI [0.24, 0.52]) but not to depressive mood (β = 0.16, ns, 95% CI [−0.01, 0.32]). Then, burnout totally mediated the relationship between techno-stressors and depressive mood (β = 0.21, *p* < 0.01, 95% CI [0.12, 0.33]), while partially mediated the relationship between techno-stressors and anxiety symptoms (β = 0.11, *p* < 0.05, 95% CI [0.05, 0.18]). The mediating effects remained statistically significant independently of the exclusion of covariates. Note that the reverse mediation model yielded a poor fit, and burnout did not mediate the reversed relationships from depressive mood and anxiety symptoms to techno-stressors, supporting the directionality of our hypothesized associations (see Table 3). Hypothesis 3a was fully supported, while Hypothesis 3b was only partially confirmed. Among the control variables, age and gender were the only statistically significant covariates. Precisely, being older was negatively related to burnout (β = −0.16; *p* < 0.05; 95% CI [−0.28, −0.03]) and depressive mood (β = −0.24; *p* < 0.01; 95% CI [−0.37, −0.11]), while being female was positively associated with depressive mood (β = 0.15; *p* < 0.05; 95% CI [0.04, 0.25]) and anxiety symptoms (β = 0.23; *p* < 0.001; 95% CI [0.12, 0.33]).

The results of the moderated mediation model (see Table 4 and Figure 2) indicated that e-work self-efficacy buffered the positive effects of techno-stressors on depressive mood and anxiety symptoms through burnout. Hence, Hypothesis 4 is confirmed. The interaction effect was negative (β = −0.15, *p* < 0.01, 95% CI [−0.24, −0.06]), suggesting that e-work self-efficacy protected against the negative effects of techno-stressors in terms of burnout and then both depressive mood and anxiety symptoms. When experiencing techno-stressors, remote workers who scored low (β = 0.15, *p* < 0.001, 95% CI [0.09, 0.22]) and moderate (β = 0.10, *p* < 0.001, 95% CI [0.06, 0.14]) on e-work self-efficacy were more likely to experience burnout and then depressive mood. Conversely, workers with high e-work self-efficacy levels did not suffer from the negative effects of techno-stressors in terms of depressive mood, as indicated by the non-significant indirect effect (β = 0.05, ns, 95% CI [0.00, 0.09]). Likewise, when exposed to techno-stressors, remote workers who scored low (β = 0.08, *p* < 0.01, 95% CI [0.03, 0.12]) and moderate (β = 0.05, *p* < 0.01, 95% CI [0.02, 0.08]) on e-work self-efficacy were more likely to develop burnout and then anxiety symptoms. The negative effects of techno-stressors in terms of anxiety symptoms were drastically lower for those who had high e-work self-efficacy levels (β = 0.02, ns, 95% CI [0.00, 0.05]). An examination of the interaction plot (see Figure 3) revealed that remote workers with high e-work self-efficacy levels experienced much lower increases in burnout compared to those with low e-work self-efficacy levels when passing from low to high conditions of techno-stressors. Additionally, the fit indices of our expected model (AIC = 13,485.92; BIC = 13,899.27) outperformed those of a competing model having resilience as an alternative moderator (AIC = 13,651.74; BIC = 14,065.09). It is also important to note that the interaction effect of resilience was not statistically significant (β = −0.07, ns, 95% CI [−0.15, 0.01]). Among the control variables, age and gender were the only statistically significant covariates. Precisely, being older was negatively related to depressive mood (β = −0.08; *p* < 0.01; 95% CI [−0.12, −0.04]), while being female was positively associated with depressive mood (β = 0.13; *p* < 0.05; 95% CI [0.04, 0.22]) and anxiety symptoms (β = 0.19; *p* < 0.001; 95% CI [0.10, 0.27]).

## 4. Discussion

The present study aimed to investigate the effects of techno-stressors on burnout and psychological health outcomes, along with the protective role of e-work self-efficacy, in a sample of remote workers. Our findings provided support for our hypotheses. Specifically, our results showed that techno-stressors were positively related to burnout and, in turn, to depressive mood and anxiety symptoms. As such, remote workers who experienced higher levels of techno-stressors were more likely to have higher levels of burnout and to present poorer psychological health. These results confirm prior research on this topic [28,47,123], showing the detrimental effect of techno-stressors on remote workers’ psychological well-being. Our results also expand the technostress literature by identifying burnout as a psychological mechanism that differentially explains how techno-stressors experienced by remote workers can be linked to depressive mood and anxiety symptoms. Thus, although burnout totally mediated the relationship between techno-stressors and depressive mood, it partially mediated the association between techno-stressors and anxiety symptoms. This suggests that techno-stressors are related to anxiety symptoms, both directly and indirectly, as mediated by burnout. Overall, these results indicated that burnout has a key role in transmitting the negative effects of (techno)stressors on psychological health outcomes [124,125], such that techno-stressed people are likely to feel exhausted or disengaged from their work, which may worsen their mood. Moreover, some ICT-related characteristics are sufficient to increase people’s anxiety levels per se, as they expose individuals to feelings of inadequacy and frustration (e.g., techno-complexity) [126], the compulsive use of technological tools, and work–life spillover (e.g., techno-invasion and tecno-overload). Notably, the reverse mediation model did not show satisfactory fit indices, supporting the hypothesized directions of the relationships from techno-stressors to mental health symptoms and not the other way around. Furthermore, our results could be better understood considering the pandemic context during which we conducted the study, which accelerated Information Technology (IT) transformations within working environments. Thus, people had to adapt quickly to the ongoing changes without having the adequate knowledge and resources to do so, in a general state of profound instability and fear [127]. In line with COR theory [30], the presence of unpredictable, multiple, and prolonged stressors may expose people to continuous resource depletion. If, in such a situation, individuals cannot counterbalance this depletion by adopting effective conservation resource strategies to restore their resources or slow down their resource losses, they are likely to experience a resource loss spiral and suffer from psychological malaise. Additionally, the present research expands the literature on the influence of the personal resources needed to effectively cope with techno-stressors. Indeed, our results showed that e-work self-efficacy buffered the relationship between techno-stressors and burnout. Hence, people with strong beliefs about their ability to handle the challenges of working remotely (such as managing activities with reduced supervision, blurred boundaries between work and private life, and virtual interpersonal interactions) were less at risk of burnout and its consequences than those with lower e-work self-efficacy levels. This result confirmed the COR theory assumptions, providing empirical evidence that people with higher resources are less vulnerable to resource loss and more able to acquire new resources. On the other hand, this protective role was not played by another personal resource, namely resilience. This finding is in line with previous studies that have attested to the stronger impact of domain-specific self-efficacy beliefs, as compared to more general self-beliefs [128] in dealing with technology-related demands [25,27], confirming its buffering effect in the stress process [92,99,129]. Since resilience represents the general capability to bounce back in the face of crisis and adversities, whereas e-work self-efficacy reflects the perceived capability to master the specific difficulties posed by remote work, we can conclude that the latter represents a more relevant coping resource for remote workers challenged by techno-stressors. Therefore, it is important to continue developing specific measures of self-efficacy beliefs that are able to intercept the varied challenges posed by new ways of working, such as remote and hybrid working. Moreover, this study contributes to the literature on the new scale proposed to assess burnout, the Burnout Assessment Tool [32], confirming its validity and distinctiveness from depression and anxiety symptoms. Finally, regarding socio-demographical variables, our results showed that older people tend to perceive a lower level of depressive mood than their younger counterparts. The literature on this relationship is mixed; some studies have found that older people are more exposed to depressive symptoms [130,131], while others have revealed a negative relationship between age and depressive mood. Our results are in line with this latter body of studies that corroborate the socio-emotional selective theory [132,133] stating that getting older can lead workers to focus on positive emotions and memories rather than on negative information, which, in turn, can contribute to maintaining their well-being. Additionally, women reported higher levels of depressive mood and anxiety symptoms, which is in line with previous research on this topic showing women’s greater susceptibility to stress and psychological health symptoms [134,135,136].

## 5. Practical Implications

From a practical perspective, our results offer relevant insights into how to promote remote workers’ psychological health. Specifically, we found that if workers have high levels of e-work self-efficacy, they will be less susceptible to experiencing negative psychological outcomes. Therefore, organizations could invest in strengthening work self-efficacy beliefs through targeted interventions, following Bandura’s recommendations and strategies [90]. In this regard, mastery experiences could be implemented by assigning remote working goals towards which employees can gradually challenge themselves within a protected setting. Likewise, verbal persuasion could be implemented by training leaders to give positive and constructive feedback and express encouragement, recognition, and trust to their employees to face the challenges of remote working. Vicarious experience could be promoted by increasing interaction, constructive comparison, and sharing of experiences among peers to facilitate the exchange of experiences and encourage the acquisition of new strategies to deal with remote working. At the same time, organizations may take steps to reduce the intensity of perceived techno-stressors, providing managers and employees with adequate training to increase their awareness of the possible risks associated with techno-related demands [137,138]. Training sessions could also be targeted to: (a) support remote workers in developing digital knowledge [139] to reduce techno-complexity; (b) manage work–life boundaries and be aware of possible spillover processes to reduce techno-invasion; (c) define realistic goals and organize realistic schedules and deadlines to reduce techno-overload; and (d) reduce the onset of burnout and psychological health symptoms among remote workers, providing training on stress management and relaxation techniques, especially targeted at younger and female employees. Moreover, remote workers could benefit from developing job resources to reduce techno-stressors. In this view, organizations could provide remote workers with the necessary technical support to handle technical difficulties [14]. Furthermore, organizations could design training programs targeted at managers to develop specific remote leadership skills, such as learning how to provide their employees with the needed resources to deal effectively with remote working.

## 6. Limitations and Future Research

The present study has some limitations that should be addressed. First, the self-rated and cross-sectional nature of our data prevented us from making causal inferences about the hypothesized relationships. However, following methodologists’ recommendations [117], we used the unmeasured method factor technique, finding that common method variance was unlikely to be a major issue in our study. Future research should collect data from multiple sources and adopt a longitudinal design. Second, we administered the survey to a small sample of remote workers from different work settings, which may not be representative of the Italian remote working population. However, given our small sample size, we adopted the parcelling technique, which allowed us to maintain an optimal indicator-to-sample-size ratio. Replications on larger and nationally representative samples of Italian workers are required to increase the generalizability of our findings. Scholars could also consider replicating our findings to compare the results obtained from workers employed in specific work sectors and cross-national samples to make comparisons. Additionally, replications should be conducted in post-pandemic times to provide a more accurate picture of the remote working experience in the “new normal” [140]. Moreover, the results of the current study might have been biased by the non-probability convenience sampling method, considering the administration of the survey through social networks. However, given its cost-effectiveness, the non-probability convenience sampling strategy has been widely used in organizational studies. Especially during the COVID-19 pandemic, this sampling method enabled scholars to reach difficult-to-track participants despite the presence of psychical distancing measures and restrictions on social interactions imposed by European governments to contain the virus [18]. Future studies should consider involving specific organizational contexts. Third, the scale referring to e-work self-efficacy was adapted from a previous contribution [25] for the purposes of the current study. Nevertheless, further in-depth scales were later developed to measure the different facets of e-work self-efficacy [27]. Therefore, future research may investigate the specific contribution of each sub-dimension as a protective factor against burnout or other outcomes (e.g., job performance [9]). Fourth, although this study extends the literature on the antecedents and consequences of burnout within a sample of remote workers, future research should address several gaps that we did not deepen. Indeed, we explored the role of a personal resource in mitigating the effect of techno-stressors on individual burnout levels, overlooking the contribution of other equally important job resources that might have acted as technostress inhibitors [8], making the experience of working remotely less stressful and more enjoyable [141]. For instance, future research may explore the extent to which remote leadership support [142] or the availability of technical support provided by the organization [143] may represent additional protective aspects or, on the other hand, facilitating factors for employees’ performance and well-being. Fifth, given that participation in this research was voluntary, selection bias cannot be ruled out. Our data might have been biased by the “healthy worker effect”, which might have led us to underestimate burnout levels, anxiety symptoms, and depressive mood [144]. Thus, it might be possible that workers who agreed to take part in our research were healthy enough to work and were overrepresented. On the contrary, burned-out workers might not have been properly represented in our study, as they might have been unable to remote work or leave the workforce due to their poor health conditions [18]. Future research could consider including an incentive for respondents to encourage everyone to participate to reduce this bias. Finally, we did not control for previous mental health states experienced by participants, which might have affected self-reports of burnout, anxiety symptoms, and depression mood, confounding the effects of techno-stressors revealed in the current study. Longitudinal replications would allow us to test whether the mental health status of remote workers varied over time as a function of techno-stressors controlling for previous mental health states.

## 7. Conclusions

The present study highlighted the process by which techno-stressors impact remote workers’ psychological health during the COVID-19 pandemic. Through a moderated mediation model, we found that burnout is a key variable in totally mediating the effects of techno-stress creators on remote workers’ depressed mood and partially mediating the effects of techno-stressors on their anxiety symptoms. Furthermore, we found that e-work self-efficacy acts as a buffer in this relationship, suggesting the importance of this personal resource in managing remote work effectively. These results provide meaningful insights into the promotion of organizational interventions aimed at empowering personal resources and protecting remote workers’ well-being.

## Figures and Tables

**Figure 1 ijerph-20-07051-f001:**
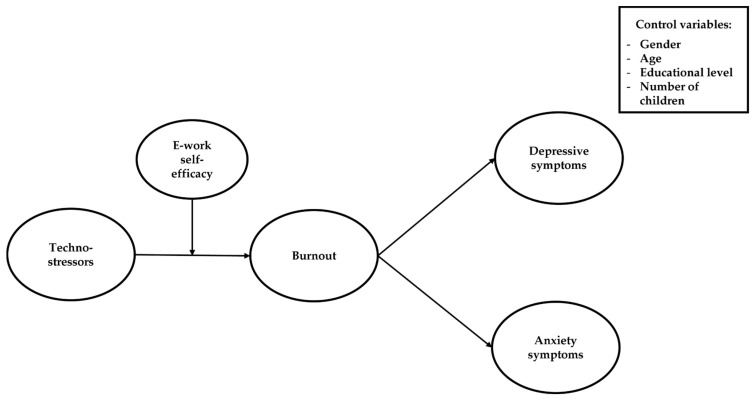
The hypothesized model of the study.

**Figure 2 ijerph-20-07051-f002:**
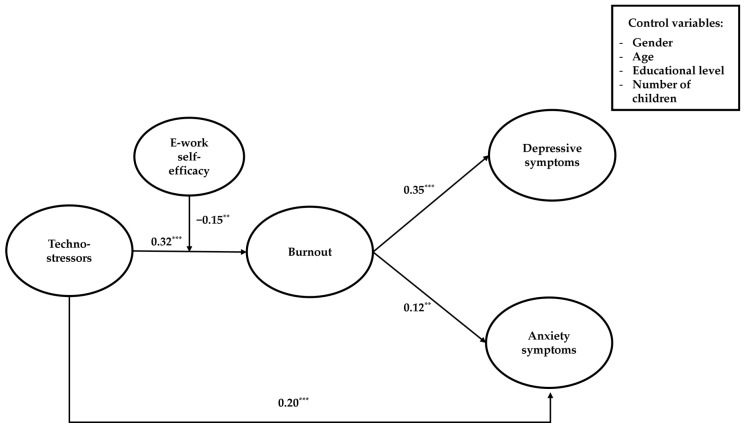
Path coefficients of the hypothesized model. *** *p* < 0.001; ** *p* < 0.01.

**Figure 3 ijerph-20-07051-f003:**
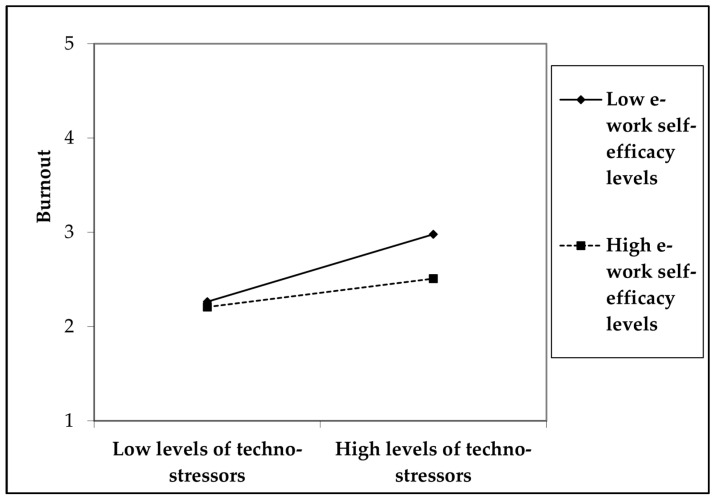
Moderating effects of e-work self-efficacy on the relationship between techno-stressors and burnout.

**Table 1 ijerph-20-07051-t001:** Intercorrelations and descriptive statistics among the study variables (*n* = 225).

	M	SD	Skew	Kurt	CR	AVE	ω	1	2	3	4	5	6	7	8	9
1. Techno-stressors	2.40	0.81	0.22	−0.54	0.94	0.55	0.94	0.94								
2. Burnout	2.24	0.63	0.38	−0.17	0.93	0.53	0.92	0.43 **^a^	0.92							
3. Depressive mood	1.78	0.49	0.58	0.26	0.87	0.57	0.81	0.29 **^a^	0.48 **^a^	0.81						
4. Anxiety symptoms	1.54	0.49	1.08	1.48	0.88	0.59	0.82	0.44 **^a^	0.36 **^a^	0.54 **^a^	0.82					
5. E-work self-efficacy	5.67	0.88	−0.74	0.26	0.85	0.52	0.77	−0.30 **^a^	−0.34 **^a^	−0.21 **^a^	−0.14 *^a^	0.77				
6. Resilience	5.02	1.16	−0.44	−0.14	0.89	0.66	0.83	−0.21 **^a^	−0.22 **^a^	−0.31 **^a^	−0.15 **^a^	0.51 **^a^	0.83			
7. Gender	-	-	-	-	-	-	-	0.13 **^b^	0.12 ^b^	0.21 **^b^	0.23 **^b^	−0.03 ^b^	−0.17 *^b^	-		
8. Age	-	-	-	-	-	-	-	0.06 ^c^	−0.09 ^c^	−0.14 **^c^	0.04 ^c^	0.13 ^*c^	0.09 ^c^	0.02 ^b^	-	
9. Education level	-	-	-	-	-	-	-	−0.05 ^c^	−0.08 ^c^	−0.05 ^c^	−0.04 ^c^	0.02 ^c^	−0.03 ^c^	0.04 ^b^	−0.09 ^b^	-
10. Children	0.77	1.01	-	-	-	-	-	0.06 ^c^	−0.06 ^c^	−0.01 ^c^	0.08 ^c^	0.12 *^c^	0.09 ^c^	0.07 ^b^	0.68 ***^b^	−0.24 **^b^

Boldfaced numbers on the diagonal represent Cronbach’s alpha; M = means; SD = standard deviations; CR = Composite Reliability; Skew. = skewness; Kurt. = kurtosis; AVE = average variance extracted; Children = number of children. * *p* < 0.05; ** *p* < 0.01; *** *p* < 0.001. ^a^ = Pearson’s correlation coefficients; ^b^ = Spearman’s rho correlation coefficients; ^c^ = Kendall’s coefficients of rank correlation tau-subb; Gender: 0 = male, 1 = female; Age: 1 = 18–25 years old, 2 = 26–35 years old, 3 = 36–45 years old, 4 = 46–55 years old, 5 = over 55 years old; Educational level: 1 = high school diploma, 2 = university degree, 3 = post-university degree; Number of children: 0 = no child, 1 = 1 child, 2 = 2 children, 3 = 3 children, 4 = more than 3 children.

**Table 2 ijerph-20-07051-t002:** Fit indices for the expected five-factor measurement model (i.e., techno-stressors, burnout, depressive mood, anxiety symptoms, and e-work self-efficacy) and the alternative models.

Model	χ^2^	df	χ^2^/df	*p*	RMSEA	90% CI RMSEA	SRMR	CFI	TLI
5 factor_cmb ^g^	721.19	364	1.98	0.00	0.06	[0.05, 0.07]	0.06	0.91	0.90
5-factor model with 2nd order factor ^f^	565.03	388	1.45	0.00	0.04	[0.04, 0.05]	0.05	0.96	0.95
5-factor model ^e^	521.34	360	1.45	0.00	0.04	[0.04, 0.05]	0.05	0.96	0.95
4-factor model ^d^	675.18	392	1.72	0.00	0.06	[0.05, 0.06]	0.06	0.93	0.92
3-factor model ^c^	1251.99	399	3.14	0.00	0.10	[0.09, 0.10]	0.09	0.79	0.77
2-factor model ^b^	1761.18	404	4.36	0.00	0.12	[0.12, 0.13]	0.10	0.67	0.64
1-factor model ^a^	2479.40	405	6.12	0.00	0.15	[0.14, 0.16]	0.12	0.49	0.45

Note. df = degree of freedom; RMSEA = Root Mean Square Error of Approximation; 90% CI RMSEA = 90% confidence interval RMSEA; SRMR = Standardized Root Mean Square Residuals; CFI = Comparative Fit Index; TLI = Tucker-Lewis Index. ^a^ All indicators load on a single factor. ^b^ Techno-stressors and e-work self-efficacy loaded on one factor; burnout, depressive mood, and anxiety symptoms loaded on a second factor. ^c^ Burnout, depressive mood, and anxiety symptoms loaded on one factor, techno-stressors loaded on a second factor, and e-work self-efficacy loaded on a third factor. ^d^ Depressive mood, and anxiety symptoms loaded on one factor, techno-stressors loaded on a second factor, and e-work self-efficacy loaded on a third factor, burnout loaded on a fourth factor. ^e^ Techno-stressors, burnout, depressive mood, anxiety symptoms, and e-work self-efficacy loaded on their respective factors. ^f^ Previous model with the inclusion of two second-order factors for techno-stressors and burnout, respectively. ^g^ Previous model with the inclusion of a common method latent variable on which makes all the items loaded.

**Table 3 ijerph-20-07051-t003:** Path coefficients and indirect effects of the model with burnout as a mediator between techno-stressors and both depressive mood and anxiety symptoms.

Model	χ^2^	df	χ^2^/df	*p*	RMSEA [90%CI]	SRMR	CFI	TLI
Expected model with controls	805.49	466	1.73	0.00	0.06 [0.05, 0.06]	0.08	0.92	0.91
Expected model without controls	634.00	366	1.73	0.00	0.06 [0.05, 0.06]	0.08	0.93	0.92
Reverse model with controls	878.39	467	1.88	0.00	0.06 [0.06, 0.07]	0.10	0.90	0.89
Reverse model without controls	634.00	366	1.73	0.00	0.06 [0.05, 0.06]	0.09	0.93	0.92
Effects					B	S.E.	95% CI	
Techno-stressors → Burnout					0.47 ***	0.08	[0.34, 0.61]	
Burnout → Depressive mood					0.44 ***	0.10	[0.28, 0.62]	
Burnout → Anxiety symptoms					0.22 **	0.08	[0.08, 0.36]	
Techno-stressors → Depressive mood					0.16	0.10	[−0.01, 0.32]	
Techno-stressors → Anxiety symptoms					0.38 ***	0.08	[0.24, 0.52]	
Gender → Techno-stressors					0.12	0.07	[0.01, 0.24]	
Age → Techno-stressors					0.03	0.10	[−0.15, 0.18]	
Educational level → Techno-stressors					−0.03	0.08	[−0.18, 0.10]	
Number of children → Techno-stressors					0.06	0.10	[−0.10, 0.23]	
Gender → Burnout					0.08	0.07	[−0.04, 0.18]	
Age → Burnout					−0.16 *	0.07	[−0.28, −0.03]	
Educational level → Burnout					−0.11	0.07	[−0.23, 0.01]	
Number of children → Burnout					−0.05	0.08	[−0.18, 0.07]	
Gender → Depressive mood					0.15 *	0.07	[0.04, 0.25]	
Age → Depressive mood					−0.24 **	0.08	[−0.37, −0.11]	
Educational level → Depressive mood					0.02	0.07	[−0.11, 0.11]	
Number of children → Depressive mood					0.15	0.08	[0.02, 0.26]	
Gender → Anxiety symptoms					0.23 ***	0.06	[0.12, 0.33]	
Age → Anxiety symptoms					−0.03	0.08	[−0.16, 0.11]	
Educational level → Anxiety symptoms					−0.01	0.07	[−0.13, 0.10]	
Number of children → Anxiety symptoms					0.07	0.08	[−0.05, 0.20]	
Techno-stressors → Burnout → Depressive mood					0.21 ***	0.06	[0.12, 0.33]	
Techno-stressors → Burnout → Anxiety symptoms					0.11 *	0.04	[0.05, 0.18]	
Total effects for depressive mood					0.37 ***	0.07	[0.25, 0.48]	
Total effects for anxiety symptoms					0.48 ***	0.07	[0.37, 0.60]	

Note. df = degree of freedom; RMSEA = Root Mean Square Error of Approximation; 90% CI RMSEA = 90% confidence interval RMSEA; SRMR = Standardized Root Mean Square Residuals; CFI = Comparative Fit Index; TLI = Tucker-Lewis Index; B = estimate; SE = standard error, 95% CI = 95% confidence interval. * *p* < 0.05; ** *p* < 0.01; *** *p* < 0.001.

**Table 4 ijerph-20-07051-t004:** Path coefficients and conditional effects for the moderated mediation model examining whether techno-stressors are related to depressive mood and anxiety symptoms through burnout, depending on e-work self-efficacy levels.

Model	AIC	BIC		
Expected model	13,485.92	13,899.27		
Alternative model with resilience	13,651.74	14,065.09		
Effects		B	S.E.	95% CI
Techno-stressors → Burnout		0.29 ***	0.06	[0.19, 0.38]
E-work self-efficacy → Burnout		−0.12	0.06	[−0.22, −0.01]
Techno-stressors * E-work self-efficacy → Burnout		−0.15 **	0.05	[−0.24, −0.06]
Burnout → Depressive mood		0.35 ***	0.07	[0.23, 0.47]
Burnout → Anxiety symptoms		0.17 **	0.06	[0.07, 0.27]
Techno-stressors → Depressive mood		0.09	0.04	[−0.01, 0.16]
Techno-stressors → Anxiety symptoms		0.20 ***	0.04	[0.13, 0.27]
Gender → Techno-stressors		0.17	0.11	[−0.01, 0.35]
Age → Techno-stressors		0.01	0.05	[−0.08, 0.10]
Educational level → Techno-stressors		−0.04	0.07	[−0.17, 0.08]
Number of children → Techno-stressors		0.04	0.07	[−0.07, 0.15]
Gender → Burnout		0.05	0.07	[−0.05, 0.16]
Age → Burnout		−0.05	0.03	[−0.10, 0.01]
Educational level → Burnout		−0.08	0.05	[−0.16, 0.01]
Number of children → Burnout		−0.01	0.04	[−0.08, 0.06]
Gender → Depressive mood		0.13 *	0.05	[0.04, 0.22]
Age → Depressive mood		−0.08 **	0.03	[−0.12, −0.04]
Educational level → Depressive mood		0.01	0.04	[−0.06, 0.06]
Number of children → Depressive mood		0.06	0.03	[0.01, 0.12]
Gender → Anxiety symptoms		0.19 ***	0.05	[0.10, 0.27]
Age → Anxiety symptoms		−0.01	0.02	[−0.05, 0.03]
Educational level → Anxiety symptoms		−0.00	0.03	[−0.06, 0.05]
Number of children → Anxiety symptoms		0.03	0.03	[−0.02, 0.08]
Techno-stressors × Low E-work self-efficacy→ Burnout → Depressive mood		0.15 ***	0.04	[0.09, 0.22]
Techno-stressors × Moderate E-work self-efficacy→ Burnout → Depressive mood		0.10 ***	0.03	[0.06, 0.14]
Techno-stressors × High E-work self-efficacy→ Burnout → Depressive mood		0.05	0.03	[0.00, 0.09]
Techno-stressors × Low E-work self-efficacy→ Burnout → Anxiety symptoms		0.08 **	0.03	[0.03, 0.12]
Techno-stressors × Moderate E-work self-efficacy→ Burnout → Anxiety symptoms		0.05 **	0.02	[0.02, 0.08]
Techno-stressors × High E-work self-efficacy→ Burnout → Anxiety symptoms		0.02	0.01	[0.00, 0.05]
Moderated indirect effect for Depressive mood		−0.05 *	0.02	[−0.09, −0.02]
Moderated indirect effect for Anxiety symptoms		−0.03 *	0.01	[−0.05, −0.01]
Total effects for Depressive mood at Low levels of E-work self-efficacy		0.24 ***	0.05	[0.16, 0.32]
Total effects for Depressive mood at Moderate levels of E-work self-efficacy		0.19 ***	0.04	[0.12, 0.26]
Total effects for Depressive mood at High levels of E-work self-efficacy		0.14 **	0.05	[0.06, 0.22]
Total effects for Anxiety symptoms at Low levels of E-work self-efficacy		0.28 ***	0.05	[0.20, 0.35]
Total effects for Anxiety symptoms at Moderate levels of E-work self-efficacy		0.25 ***	0.04	[0.18, 0.32]
Total effects for Anxiety symptoms at High levels of E-work self-efficacy		0.22 ***	0.04	[0.15, 0.30]

Note. AIC = Akaike Information Criterion; BIC = Bayesian Information Criterion; SE = standard error, 95% CI = 95% confidence interval. * *p* < 0.05; ** *p* < 0.01; *** *p* < 0.001.

## Data Availability

The data that support the findings of this study are available from the corresponding author upon reasonable request.

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
