# Peer review of "Techno-Stress Creators, Burnout and Psychological Health among Remote Workers during the Pandemic: The Moderating Role of E-Work Self-Efficacy"

_ijerph, 2023, doi:10.3390/ijerph20227051_

Round 1

Reviewer 1 Report

Comments and Suggestions for Authors

Dear Author,

Topic is interesting. Remote workers is interesting population for research. You have great effort in your research, but there is some several points, and article could be improved.

Methodology: Sample is quite small. Participants are from different settings and they were invited to participate in the study from the social network and they answered  through the e-platform.

Hypothesis (2-4) are not something new. Results (tables) are not clear enough. Conclusion is missing.

I hope you could publish your results after rewriting the paper.

Best wishes

Author Response

RESPONSES TO REVIEWER’S COMMENTS

(Legend: R1.1 stands for Reviewer 1, comment 1, and A1.1 for our reply to this comment)

R1.0

Dear Author,

Topic is interesting. Remote workers is interesting population for research. You have great effort in your research, but there is some several points, and article could be improved.

A1.0

Thank you for appreciating our work and providing us with useful comments that helped us improve the quality of our manuscript.

R1.1

Methodology: Sample is quite small. Participants are from different settings and they were invited to participate in the study from the social network and they answered through the e-platform.

A1.1

Thank you for raising this important point. We are aware that these points represent potential limitations of our study. However, our research was conducted in the first year of the Covid19 pandemic, in which collecting data was extremely challenging, especially in Italy. Therefore, in this version we tried to better explain our methodological choices. In fact, given our small sample size, to keep statistical power, we created item parcels, so that each construct was measured by three items (i.e., 3 items for techno-stressors, 3 for burnout, 3 for depressive mood, 3 for anxiety symptoms, 3 for e-work self-efficacy for a total of 15 items). This enabled us to have an optimal indicator-to-sample size ratio. Thus, since the recommended ratio is 10 subjects for 1 item, our sample size was appropriate (i.e., a minimum of 150 subjects). Regarding data collection, while in our view the employment of an internet platform can be considered appropriate for remote workers, the use of social networks may expose to the risk of selection bias. This choice was made to maximize data collection in light of ongoing COVID-19-containment in the Italian context. Accordingly, we have extended our “Limitations and future research” section by adding the following paragraph (please see pp. 19/20):

“Second, we administered the survey to a small sample of remote workers from different work settings and that may be not representative of the Italian remote working population. However, given our small sample size, we adopted the parcelling technique, which al-lowed us to maintain an optimal indicator-to-sample-size ratio. Replications on larger and nationally representative samples of Italian workers are required to increase the generalizability of our findings. Scholars could also consider replicating our findings to com-pare the results obtained from workers employed in specific work sectors and cross-national samples to make comparisons. Additionally, replications should be conducted in post-pandemic times to provide a more accurate picture of the remote working experience in the “new normal” [140]. Moreover, the results of the current study might have been biased by the non-probability convenience sampling method considering the administration of the survey through social networks. However, given its cost-effectiveness, the non-probability convenience sampling strategy has been widely used in organizational studies. Especially during the COVID-19 pandemic, this sampling method enabled scholars to reach difficult-to-track participants despite the presence of psychical distancing measures and restrictions to social interactions imposed by European governments to contain the virus [18]. Future studies should consider involving specific organizational contexts.”

R1.2

Hypothesis (2-4) are not something new.

A1.2

Thank your valuable feedback. We agree with you that the relationship between burnout and mental health outcomes has been well-studied, and the protective role of self-efficacy in this relationship is well-proven. Nevertheless, we believe that our study offers some new elements that need to be considered.

First, the great majority of burnout studies employed the Maslach Burnout Inventory to measure burnout, which consider it as the result of three different dimensions (namely Exhaustion, Cynicism, and reduced Efficacy) that cannot be combined. Consequently, often the role of the key burnout dimension, namely Exhaustion, was overemphasized (e.g., Bosmans et al., 2019), underestimating the other burnout dimensions (Androulakis et al., 2023).

On the contrary, in our study we used the Burnout Assessment Tool (BAT-12), which allow to combine burnout dimensions in a single burnout total score, and which includes two new dimensions, namely emotional and cognitive impairment, in addition to exhaustion and cynicism (now labelled mental distance). Although there is some preliminary evidence of correlations between BAT and mental health, specifically depression (Bianchi, Cavalcante, et al., 2023), this is the first study to empirically investigate whether burnout, as measured by the BAT, can be a mechanism for translating the effects of techno-stressors in anxiety and depressive symptoms.  Accordingly, we have better explained in the introduction how our study contributes to fill the gap in the existing literature by underlying its main novelties in the literature review and in the assumptions related the hypotheses (pp. 4-6):

“As a result, most studies using the MBI to measure burnout consider it the product of three dimensions that cannot be combined. Consequently, often the role of the key dimension of burnout, namely exhaustion, has been overemphasized [e.g., 62], underestimating the other burnout dimensions [e.g., 62, 63]

(…)

“Based on the Burnout Assessment Tool [32], burnout represents a multifaceted experience, where exhaustion and disengagement from one's work are compounded by the weakening of control over cognitive and emotional processes (e.g., having trouble focusing or controlling one's negative emotional states at work), thus making the person more susceptible to developing physical and psychological problems [32]

(…)

“However, most studies on the relationships of burnout with depressive mood and anxiety symptoms analysed burnout as measured by the MBI, mainly focusing on the emotional exhaustion dimension, and underestimating the other burnout components. To date, only a very few studies showed positive correlations of burnout as measured by the BAT, with depressive mood [74] and anxiety symptoms [75], with no previous research on remote workers during the pandemic..”

(…)

“Although there is some preliminary evidence of correlations between BAT and mental health symptoms (including depression and anxiety symptoms [74,75]), this is the first study to empirically investigate whether the total burnout score, as measured by the BAT, can be a mechanism for translating the effects of techno-stressors in burnout and depressive symptoms.”

(…)

“Specifically, the dysregulation of cognitive and emotional processes that characterize the burnout syndrome, as it is conceptualized according to the Burnout Assessment Tool [32], may imply an increase in negative thoughts, worries, and rumination, resulting in a worsening of mood and an increase in anxiety symptoms [76, 77].”

Moreover, with respect to the moderation hypothesis of self-efficacy, this study focuses on a construct related specifically to the population of remote workers (i.e., e-work self-efficacy) that differs from others previously explored (e.g., computer self-efficacy). Accordingly, we have extended our hypothesis formulation by underlying the importance of this personal resource and the novelty derived by analysing this construct. Now you read as follows (see p. 2):

“For this reason, perceiving oneself as being able to handle the challenges associated with working remotely (e.g., managing work and private boundaries appropriately, maintaining a positive working relationship with colleagues and the manager while working remotely) represents a valuable resource in this new working scenario. Indeed, self-efficacy in the context of remote work (i.e., e-work self-efficacy [25]) enables workers to better cope with techno-stressors and then protect them from developing job stress-related detrimental outcomes [16, 26-28]. Although the protective role of self-efficacy against burnout and mental health problems is well-known [29], this is the first study to investigate e-work self-efficacy as a personal boundary condition explaining when the effects of techno-stressors can translate into remote workers’ burnout and mental health problems. Investigating the protective role of e-work self-efficacy is important because it provides new practical in-sights into how practitioners can support remote workers’ mental health by providing tailored training programs on the development of their digital resilience.”

R1.3

Results (tables) are not clear enough.

A1.3

Thank you for your helpful comment. To further clarify the content of our Tables, we changed their titles as follows:

“Table 2. Fit indices for the expected five-factor measurement model (i.e., techno-stressors, burnout, depressive mood, anxiety symptoms, and e-work self-efficacy) and the alternative models.”

“Table 3. Path coefficients and indirect effects of the model having burnout as a mediator between techno-stressors and both depressive mood and anxiety symptoms.”

“Table 4. Path coefficients and conditional effects for the moderated mediation model examining whether techno-stressors are related to depressive mood and anxiety symptoms through burnout depending on e-work self-efficacy levels.”

Please note that the content of the tables is reported in line with what has been done in the following previous studies published in IJERPH:

  • Maffoni, M., Fiabane, E., Setti, I., Martelli, S., Pistarini, C., & Sommovigo, V. (2022). Moral Distress among Frontline Physicians and Nurses in the Early Phase of COVID-19 Pandemic in Italy. International Journal of Environmental Research and Public Health19(15), 9682.
  • Rego, F., Sommovigo, V., Setti, I., Giardini, A., Alves, E., Morgado, J., & Maffoni, M. (2022). How supportive ethical relationships are negatively related to palliative care professionals’ negative affectivity and moral distress: A Portuguese sample. International journal of environmental research and public health19(7), 3863.
  • Sommovigo, V., Bernuzzi, C., Finstad, G. L., Setti, I., Gabanelli, P., Giorgi, G., & Fiabane, E. (2023). How and When May Technostress Impact Workers’ Psycho-Physical Health and Work-Family Interface? A Study during the COVID-19 Pandemic in Italy. International Journal of Environmental Research and Public Health20(2), 1266.

R1.4

Conclusion is missing.

A1.4

As recommended, we have expanded the conclusions section in our revised version of the manuscript. Now you read as follows (see pp.):

“The present study highlighted the process by which techno-stressors impact remote workers' psychological health during the Covid-19 pandemic. Through a moderated mediation model, we found that burnout is a key variable in totally mediating the effects of techno-stress creators on remote workers’ depressed mood and partially mediating the effects of techno-stressors on their anxiety symptoms. Furthermore, we found that e-work self-efficacy acts as a buffer in this relationship, suggesting the importance of this personal resource in managing remote working effectively. These results provide meaningful insights into the promotion of organizational interventions aimed at empowering personal resources and protecting remote workers’ well-being.”

R1.5

I hope you could publish your results after rewriting the paper.

Best wishes

A1.5

Thank you very much for your helpful comments. We have incorporated in our revised version of the manuscript all the edits that you have thoughtfully offered us.  We hope that you will appreciate our revised manuscript.

Reviewer 2 Report

Comments and Suggestions for Authors

I thank the authors for courageously writing about such an interesting topic.

I recommend addressing the following aspects:

1.       The summary should indicate the tests used and the type of methodology used.

2.       Improve the formulation of objectives. Define objectives without justifying why. This should be done in your theoretical framework. Simplify the formulation of the three main objectives of the study

3.       The ethical aspects of the study should be clarified. Name of the ethics committee that approved the study, and date. How did the authors guarantee the confidentiality of the data? Was the Declaration of Helsinki considered in the sampling and data collection process?El muestreo por conveniencia no es un tipo de muestreo que garantice la fiabilidad de los resultados.

States such as the mental health or depressive state of the participants were not taken into account as inclusion criteria to ensure an adequate distribution of the sample. Was the distribution of the sample normal distribution, considering the mental health states of the participants?

In the objectives of the study, the researchers raise questions about the modulation of mental health variables such as depression, anxiety, and stress. How did the researchers control for these antecedent states in the participants?

Review the entire document and cite appropriately following the journal's guidelines. For example: ....Techno-stressors were adapted by the scale by Ragu-Nathan et al., [99],…. Remove authors' names from the entire manuscript

Comments on the Quality of English Language

The manuscript must be edited by a native speaker

Author Response

RESPONSES TO REVIEWER’S COMMENTS

(Legend: R2.1 stands for Reviewer 2, comment 1, and A2.1 for our reply to this comment)

R2.0

I thank the authors for courageously writing about such an interesting topic.

A2.0

We are glad that you find the topic of our study interesting.

R2.1

I recommend addressing the following aspects:

  1. The summary should indicate the tests used and the type of methodology used.

A2.1

Thank you for your constructive comment. Accordingly, we have added the following a sentence in our revised Abstract (please see p. 1):

“The data were analyzed using structural equation mediation and moderated mediation models adopting a parceling technique.”

R2.2

  1. Improve the formulation of objectives. Define objectives without justifying why. This should be done in your theoretical framework. Simplify the formulation of the three main objectives of the study

A2.2

Thank you for raising this important point. We have reworded the objectives of our study in order to simplify (and clarify) their formulation. Now you read as follows (please see p. 3):

“Accordingly, the aims of the present research are threefold:

1) to explore whether techno-stressors are related to burnout, measured using the Burnout Assessment Tool [32];

2) to examine whether burnout mediates the relationship between techno-stressors and psychological health outcomes (i.e., anxiety symptoms and depressive moods);

3)  to investigate whether the detrimental effects of techno-stressors on anxiety symptoms and depressive mood through burnout are buffered by e-work self-efficacy levels.

R2.3

  1. The ethical aspects of the study should be clarified. Name of the ethics committee that approved the study, and date. How did the authors guarantee the confidentiality of the data? Was the Declaration of Helsinki considered in the sampling and data collection process?

A2.3

Thank you for your important comment. In fact, for our study it was not necessary to have the Ethical Board Approval. In fact, according to national legislation, this is a non-interventional study which involved less than minimal risk. All procedures performed in this study were in accordance with the ethical standards of the national research committee and with the 1964 Helsinki declaration and its later amendments or comparable ethical standards. All participants provided their signed informed consent before participating in the study and data storage met all current Data Protection regulations (GDPR). We clarified in the text these important aspects.

“All procedures performed in this study were in accordance with the ethical standards of the 1964 Helsinki declaration and its later amendments or comparable ethical standards. Data storage met current Data Protection regulations. The questionnaire’s cover sheet in-formed participants about the study’s goal and ensured both the voluntariness of their participation and the confidentiality of the responses.”

R2.4

  1. El muestreo porconveniencia no es un tipo de muestreo que garantice lafiabilidad de los resultados.

A2.4

Thank you for raising this important point. We are aware of the limitations of our convenience sample, however during the pandemic it was very difficult to contact potential participants, therefore we decided to use a non-probability convenience sampling strategy using social network. However, we clearly mentioned this aspect in the limitation of the study (see page 19):

“Moreover, the results of the current study might have been biased by the non-probability convenience sampling method considering the administration of the survey through social networks. However, given its cost-effectiveness, the non-probability convenience sampling strategy has been widely used in organizational studies. Especially during the COVID-19 pandemic, this sampling method enabled scholars to reach difficult-to-track participants despite the presence of psychical distancing measures and restrictions to social interactions imposed by European governments to contain the virus [18]. Future studies should consider involving specific organizational contexts.”

R2.5

States such as the mental health or depressive state of the participants were not taken into account as inclusion criteria to ensure an adequate distribution of the sample. Was the distribution of the sample normal distribution, considering the mental health states of the participants?

A2.5

As suggested by kurtosis and skewness values, the distribution of our sample was normal, considering the mental health status of our participants. No outliers were detected or deleted. However, selection bias cannot be excluded. Therefore, we explicitly mention this aspect and possible influence of well-known healthy worker effect among our study limitations:

“Fifth, given that the participation in this research was voluntary, selection bias cannot be ruled out. Our data might have been biased by the “healthy worker effect”, which might have led us to underestimate burnout levels, anxiety symptoms, and depressive mood [144]. Thus, it might be possible that workers who agreed to take part in our research were healthy enough to work and then were overrepresented. On the contrary, burned-out workers might not have been properly represented in our study as they might have been unable to remote work or leave the workforce due to their poor health conditions [18]. Future research could consider including an incentive for respondents to encourage everyone to participate to reduce this bias”.

R2.6

In the objectives of the study, the researchers raise questions about the modulation of mental health variables such as depression, anxiety, and stress. How did the researchers control for these antecedent states in the participants?

A2.6

Thank you for your constructive feedback. Unfortunately, we could not control for previous mental health states of participants. We have extended our Limitations section by adding the following paragraph (see p. 20):

“Finally, we did not control for previous mental health states experienced by participants, which might have affected self-reports of burnout, anxiety symptoms, and depression mood confounding the effects of techno-stressors revealed in the current study. Longitudinal replications would allow us to test whether the mental health status of remote workers varied over time as a function of techno-stressors controlling for previous mental health states.”

R2.7

Review the entire document and cite appropriately following the journal's guidelines. For example: Techno-stressors were adapted by the scale by Ragu-Nathan et al., [99],…. Remove authors' names from the entire manuscript.

A2.7

Thank you for your helpful feedback. We remove all authors names from the manuscript, and we accurately revise all the references following journal guidelines.

R2.8

The manuscript must be edited by a native speaker

A2.8

Thank you very much. We carefully revised the entire manuscript for any typo. In addition, we have had the article reviewed by a native English speaker.  We hope that you will appreciate our revised manuscript.

Round 2

Reviewer 2 Report

Comments and Suggestions for Authors

The authors have introduced and improved the suggested changes. I think the manuscript can be considered for publication

Kind regards

Comments on the Quality of English Language

Minor editing of English language required

Author Response

RESPONSES TO REVIEWER 2’S COMMENTS

(Legend: R2.1 stands for Reviewer 2, comment 1, and A2.1 for our reply to this comment)

R2.0

The authors have introduced and improved the suggested changes. I think the manuscript can be considered for publication.

A2.0

Thank you very much for appreciating the efforts made to review our manuscript and endorsing the publication of our manuscript.

R2.1

Minor editing of English language required.

A2.1

We proofread the manuscript and checked the English language and style using grammar editing software (i.e., Grammarly). We would like to thank you again for your constructive comments that helped us improve the quality of our manuscript.